# The Promotion of Migration and Myogenic Differentiation in Skeletal Muscle Cells by Quercetin and Underlying Mechanisms

**DOI:** 10.3390/nu14194106

**Published:** 2022-10-02

**Authors:** Tzyh-Chyuan Hour, Thi Cam Tien Vo, Chih-Pin Chuu, Hsi-Wen Chang, Ying-Fang Su, Chung-Hwan Chen, Yu-Kuei Chen

**Affiliations:** 1Department of Biochemistry, School of Medicine, Kaohsiung Medical University, Kaohsiung 80708, Taiwan; 2Department of Medical Research, Kaohsiung Medical University Hospital, Kaohsiung 80708, Taiwan; 3Institute of Cellular and System Medicine, National Health Research Institutes, Miaoli 350401, Taiwan; 4Orthopaedic Research Center, Kaohsiung Medical University, Kaohsiung 80708, Taiwan; 5Department of Orthopedics, College of Medicine, Kaohsiung Medical University, Kaohsiung 80708, Taiwan; 6Division of Adult Reconstruction Surgery, Department of Orthopedics, Kaohsiung Medical University Hospital, Kaohsiung Medical University, Kaohsiung 80708, Taiwan; 7Department of Orthopedics, Kaohsiung Municipal Ta-Tung Hospital, Kaohsiung Medical University, Kaohsiung 80708, Taiwan; 8Regeneration Medicine and Cell Therapy Research Center, Kaohsiung Medical University, Kaohsiung 80708, Taiwan; 9Musculoskeletal Regeneration Research Center, Kaohsiung Medical University, Kaohsiung 80708, Taiwan; 10Department of Food Science and Nutrition, Meiho University, Pingtung 91202, Taiwan

**Keywords:** quercetin, myogenic differentiation, migration, sarcopenia

## Abstract

Aging and muscle disorders frequently cause a decrease in myoblast migration and differentiation, leading to losses in skeletal muscle function and regeneration. Several studies have reported that natural flavonoids can stimulate muscle development. Quercetin, one such flavonoid found in many vegetables and fruits, has been used to promote muscle development. In this study, we investigated the effect of quercetin on migration and differentiation, two processes critical to muscle regeneration. We found that quercetin induced the migration and differentiation of mouse C2C12 cells. These results indicated quercetin could induce myogenic differentiation at the early stage through activated p-IGF-1R. The molecular mechanisms of quercetin include the promotion of myogenic differentiation via activated transcription factors STAT3 and the AKT signaling pathway. In addition, we demonstrated that AKT activation is required for quercetin induction of myogenic differentiation to occur. In addition, quercetin was found to promote myoblast migration by regulating the ITGB1 signaling pathway and activating phosphorylation of FAK and paxillin. In conclusion, quercetin can potentially be used to induce migration and differentiation and thus improve muscle regeneration.

## 1. Introduction 

Sarcopenia is the progressive loss in muscle mass, muscle strength, and function most often related to aging [1]. This decline in muscle function and muscle mass can lead to reduced mobility, worsened quality of life, increased hospitalizations stemming from fall injuries, and longer rehabilitation times [1]. One important aspect of sarcopenia is the reduced ability of aged muscles to regenerate. Healthy skeletal muscle tissues have the ability to regenerate in response to mechanical stress and heal small injuries. This regeneration is a complex and highly coordinated process involving myoblast migration and differentiation [2]. Myoblasts migrate to the site of muscle damage, which is a critical step to myoblast alignment and fusion into multinucleated myotubes [3]. Myogenic differentiation initiates, and myoblast permanently exits the cell cycle, changing morphology, fusing into multinucleated myotubes, and expressing a muscle-specific protein myosin heavy chain (MHC) [4].

Myoblasts need to migrate through the extracellular matrix (ECM) to reach the site of injury during repair [5]. The cell migration process requires some coordination of certain processes, including rapid changes in an actin filament’s dynamics undergoing rapid changes, as well as assembling and disassembling at adhesion sites. Cell migration and adhesion are affected by focal adhesions formed by the extracellular matrix, integrins, and the cellular cytoskeleton at cell junctions [6]. Integrin, which is a heterodimeric transmembrane receptor composed of alpha and beta subunits, plays an important role in the conversion of extracellular signals into intracellular responses and influences extracellular matrix responses. Although there is a large number of integrin receptor complexes, skeletal muscle integrin receptors are limited to seven alpha subunits, all associated with the beta1 integrin subunit [7]. The activation of focal adhesion kinase (FAK), a nonreceptor tyrosine kinase localized at focal adhesions (FA), is required for integrin signal transmission [7]. Paxillin is a major component of FA and is responsible for mediating the transduction of extracellular signals to intracellular signals, a process triggered by the interaction of the ECM with integrins [8]. One recent study using a bovine model found that SPARCL1, a component of ECM2, binds to integrin β1 via mediated overexpression of phosphorylated FAK and paxillin to promote skeletal muscle satellite cell migration [9].

Growth factors play a vital role in triggering myogenic differentiation and myotube fusion during skeletal muscle regeneration. Platelet-derived growth factors (PDGF) control a wide variety of cellular reactions, including cell migration, cell differentiation, and tissue remodeling [10]. PDGFs have five dimer isoforms, including PDGF-AA, PDGF-BB, PDGF-AB, PDGF-CC, and PDGF-DD, while PDGF receptors have two isoforms, PDGFRα and PDGFRβ, which can bind to five distinct PDFG ligands. PDGFRα is activated by PDGF-AA and CC, while PDGFRβ is activated by PDGF-BB and DD [11]. A cell population expressed by PDGFRβ has been found among perivascular cells that become pluripotent stem cells within skeletal muscle tissue [12]. In response to injury, PDGF-BB is released from injured myofibers, causing platelets to degranulate and macrophages to invade, promoting satellite cell migration, increasing satellite cell proliferation during skeletal muscle regeneration while inhibiting satellite cell differentiation [13].

Insulin-like growth factor 1 (IGF-1) is a growth factor peptide that medicates autocrine and paracrine signaling in skeletal muscle and has enabled repair of muscle damage through its effect on muscle differentiation and proliferation [14]. IGF-1 regulates the phosphoinositide 3-kinase (PI3K)/protein kinase B (AKT), the primary downstream target of IGF-1, promoting skeletal muscle differentiation and protein synthesis [15]. Activation of AKT regulates diverse downstream signaling, including mTOR/4EBP1/p70S6K and GSK-3β/β-catenin [15]. Similarly, IGF-1 has been found to stimulate tyrosine-phosphorylated STAT3 in C2C12 muscles and in different embryonic and adult mouse organs during different developmental stages [16]. Moreover, the activation of STAT3 significantly enhances myogenic differentiation, and the inhibition of STAT3 suppresses transcription factors MEF2 and myogenin, inhibiting myogenic differentiation [4].

During their differentiation into myotubes, myoblasts increase ATP activity levels and energy requirements [17]. MHC genes are expressed in response to activated AMPK, which phosphorylates and retains class IIA HDACs outside the myonucleus [18]. A previous study found that the level of AMPK phosphorylation (Thr172), possibly indicating variation in energy, changes in response to exercise, and AICAR has been found in vitro to activate AMPK and stimulate the differentiation of skeletal muscle satellite cells and found in vivo to induce the myogenic formation of satellite cells [19].

Quercetin (3,3′,4′5,7-Pentahydroxyflavone, C_15_H_10_O_7_), a natural flavonoid, has been detected in many vegetables and fruits, including onions, apples, berries, tomatoes, seeds, and nuts, as well as in many leaves, flowers, barks, and tea [20]. Several studies have demonstrated the beneficial health benefits of quercetin, which include reducing inflammatory and oxidative stress, which enhances athletic performance, protecting against cell injury, reducing blood cholesterol, and having anticarcinogenic effects in several cell cultures and animal models [21]. Quercetin has also been found to protect against muscle atrophy in several situations, including cachexia, obesity, and muscle disuse [21,22]. However, while its effects have been studied, there has been no study exploring its effect on myoblast migration and myogenic differentiation, the mechanisms through which it may exert its effect. Therefore, in this study, we investigated the effect of quercetin on myoblast migration and myogenic differentiation by studying its effect on the ITGB1 signaling pathway and related to downstream signaling of the IGF-1R pathway. We also investigated the effect of quercetin at the early stage of myogenic differentiation.

## 2. Materials and Methods

### 2.1. Cell Line and Reagents

Mouse skeletal muscle C2C12 myoblast cell line (Number: CRL-1772) was purchased from American Type Culture Collector (ATCC, Manassas, VA, USA). Dulbecco’s modified Eagle’s medium (DMEM) was obtained from Thermo Scientific (Rockford, IL, USA). Fetal bovine serum (FBS), horse serum, penicillin/streptomycin, and L-glutamine were acquired from Gibco (Waltham, MA, USA). Antibodies against MHC (SC-376157), AKT (SC-1619), p-paxillin (Tyr118, Sc-101774), and FAK (#I2112) were purchased from Santa Cruz (Dallas, TX, USA). IGF-1R (#3027), p-IGF-1R (#3024S), PDGFR-β (#4564), p-PDGFR- β (#3124), p-AKT(T308) (#2965S), p-AKT(Ser473) (#4060S), AMPK (#5832S), p-AMPK (T172) (#2535), GSK-3β (#12456), p-GSK-3β (Ser 9) (#5558), β-catenin (#8480), mTOR (#2983), p-mTOR(S2448) (#5536), 4EBP1 (#9644), p-4EBP1 (Thr37/46) (#2855), P70S6K (#2708), p-P70S6K (#9234), p-STAT3 (#9131S), STAT3 (#4060S), ITGB1 (#9699), and p-FAK (#3283S) were purchased from Cell Signaling (Danvers, MA, USA). Paxillin was obtained from BD Biosciences, while GAPDH and β-actin were purchased from GeneTex (Alton, CA, USA). LY294002 (AKT inhibitor) was purchased from Cayman Chemical (#Cat79020). Quercetin (HPLC ≥95%, Sigma, Milpitas, CA, USA) was soluble in dimethyl sulfoxide (DMSO) solution. DMSO is a toxic solvent for cell growth; therefore, the quercetin stock solution was soluble in a growth medium or differentiation medium containing 0.2% DMSO. Thus, 0.2% DMSO must be added to the total volume medium of the control group for each experiment needed. 

### 2.2. Cell Differentiation and Migration

C2C12 myoblast cells were seeded 5 × 10^4^ in a 24 well-plate with 800 μL DMEM medium (per a well) containing 10% FBS, 1% L-glutamine, and 1% penicillin/streptomycin (known as a growth medium, GM) to induce cell proliferation until 80% confluency. C2C12 myoblast cells were stimulated to cell differentiation and fusion into myotubes by replacing the original medium with differentiation medium (DM) containing DMEM and 2% horse serum (HS) (Day 0). To complete the myoblast cell differentiation process, the cells were treated with different concentrations of quercetin (0, 2.5, 12.5, 25, 50 μM) for 7 days every 48 h, at which point the medium was replaced with a fresh new drug and differentiation medium. To promote cell migration, C2C12 cells were grown to 80% confluency density in a 24-well plate, each well treated with quercetin 0, 12.5 µM and cultured in a DMEM medium containing 2% horse serum for 15 h.

### 2.3. Cell Viability Assay (MTT Assay)

C2C12 myoblast cells were seeded at 24 h at a rate of 1200 cells per well in a 96-well plate, each well containing 100 μL growth medium for whole-cell adhesion and growth to about 40% density. C2C12 cells were then treated with different concentrations of quercetin (0, 5, 50, 100, 200, 300, 400, 500 μM) for 72 h to a final medium volume of 200 μL DMEM. After treatment, C2C12 cells in each well were added with 50 μL (2 mg/mL) of the MTT drug. Afterwards, the 96-well plate was incubated for 4 h in a humidified atmosphere at 37 °C, of 95% air and 5% CO_2_. Then, the well plate was centrifuged at 1000 rpm for 10 min to ensure stable cell adhesion in the plate. The medium was then carefully discarded. In all, 150 μL DMSO water-insoluble formazan was added into each well. Absorbance wavelength of the formazan product was measured under 540 nm using a micro-plate (ELISA) reader.

### 2.4. Immunofluorescence Staining and Morphology Analysis

C2C12 cells were fixed with 4% paraformaldehyde for 20 min at room temperature to preserve cell morphology. Then, they were washed three times by PBS followed by permeabilizing with 0.2% Triton X-100. To prevent non-specific antibody binding, C2C12 cells were blocked with PBS solution containing 1% bovine serum albumin (BSA) and 5% normal goat serum (NGS) for 1 h. C2C12 cells were then incubated with MHC primary antibody diluted in blocking buffer (1:250) and maintained at 4 °C overnight. The following day, after being washed five times using PBS, the C2C12 cells were incubated with anti-mouse secondary antibody (1:1500) for 1 h at room temperature. The nuclei of myotubes were then stained with 4’,6-diamidino-2-phenylindole (DAPI) for 10 min. They were again washed five times using PBS before being observed under fluorescence microscope (LEICA) 10× magnification to capture C2C12 myotube morphology. Three images taken using different fields of view for each well of each control well and each quercetin treated were randomly selected. The percent fusion index was calculated by the total nucleic within the MHC-positive myotube (≥2) divided by the total number nuclei counted in the image field. The number of nuclei was counted in Image J software using the following formula.
$$Fusion\;index\;\left(\%\right)=\frac{total\;number\;of\;nucleic\;in\;myotubes}{total\;number\;of\;nucleic\;in\;DAPI\;positive\;cells}\times 100$$

### 2.5. Myoblast Cell Analysis and Flow Cytometry

C2C12 myoblast cells were seeded 6 × 10^5^ into a small (6cm) plate with a 2 mL growth medium (GM) until the cells were grown to 80% density. To synchronize cells, we replaced the medium C2C12 cells were in with a low-serum DMEM medium containing 2% FBS, 1% L-glutamine, and 1% penicillin/streptomycin and left the cultures to incubate for 24 h. Then, the medium C2C12 cells were in was replaced with a DMEM medium containing 2% horse serum (control group) and treated with quercetin (12.5 μM) for 6, 12, and 24 h. After each time point, cells were collected and fixed with 70% cooling-ethanol overnight. Cells were incubated with 0.1% Triton X-100, RNase (0.2 mg/mL) under 37 °C and 5% CO_2_ for 1 h. Their DNA was stained using Propidium Iodide (PI, 2 mg/mL)) and left to sit for 30 min in a dark room at room temperature. The cell cycle phase was analyzed using Attune N × T cytometer software.

### 2.6. Wound-Healing Assay

C2C12 myoblast cells were seeded at 5 × 10^4^ in a 24 well-plate with 800 μL GM (per well). When cells were grown to 80% density and removed the GM medium, the C2C12 cells were scratched in the center of the well with a 200 μL tip. Then, after being washed twice with PBS solution to eliminate cell fragmentation, C2C12 myoblast cells were assigned to place in into 500 μL DM alone (control group) to be treated with quercetin (study group). Using a microscope (LEICA) with 4× magnification, we captured images of cells migration to the scratched area. Wound areas were measured using ImageJ software.

### 2.7. Transwell Migration Assay

Transwell migration assay was performed in a 24-well plate using membranes with pore sizes equaling 8 μM. The cells were seeded 2 × 10^4^ in the upper chamber with 250 μL serum-free DMEM medium and in the lower chamber with 750 μL DMEM + 2% horse serum and left to incubate under 37 °C and 5% CO_2_ for 15 h. After 15 h, the cells that had migrated into the lower chamber were fixed with 3.7% paraformaldehyde for 5 min, permeabilized with 100% methanol for 25 min, and stained with 0.2% crystal violet for 5 min. The cells that had not migrated were removed by washing twice with PBS and a cotton swab. An image of the cell migration that had taken place was captured by microscope (LEICA) under 4X magnification. The migrated cells stained with crystal violet fluorescence were removed with 0.1% acetic acid, and absorbance was measured at 595 nm.

### 2.8. AKT Inhibitor Treatments

C2C12 cells were seeded at 5 × 10^4^ in 24-well plates. Once the cells had grown to 80% density, their medium was replaced with DMEM medium containing 2% horse serum to induce cell differentiation. To test the effect of quercetin reserve on AKT inhibitor (LY294002) inhibited cell differentiation, we first treated the C2C12 cells with 10 μM of LY294002 for 30 min and then treated them with 12.5 μM of quercetin for 7 days, replacing the medium and drug every 48 h.

### 2.9. Western Blot Analysis

C2C12 cells were collected after quercetin treatment period times. We placed them in RIPA buffer containing 1% protease inhibitor and 1% phosphatase inhibitor for cell lysis and centrifuged them at 13,000 rpm under 4 °C for 30 min to extract protein. The protein concentration of the sample was measured by BCA assay. In all, 30 μg protein concentration of sample was loaded on 8–15% SDS–PAGE in a vertical electrophoresis chamber set at 100 V–110 V, 400 mA, for 210–240 min. The PVDF membrane was cut to fit the size of the gel and then the PVDF membrane was incubated for 5 min in methanol to activate it. The sandwich was placed in a transfer tank filled with transfer buffer and run at 100V, 400 mA, for 70–90 min. The membrane was then incubated in a TBST solution containing 5% skim milk for 1 h followed by incubation with primary antibody under 4 °C overnight. The next day, the PVDF membrane was washed three times and incubated with secondary antibodies for 1 h. The PVDF membranes were stained with ECL fluorescence (1:1) solution and analyzed using multiGel-21 (MGSI-21-C2-1M, 2M) CCD digital imaging to detect protein.

### 2.10. Statistical Analysis

All results experiments were replicated at least three times. The data are presented as the mean and standard deviation ± SD. All statistical operations were performed using Sigma Software. ANOVA with Tukey HSD was used to assess significance, set at *p* < 0.05. Image J software was used for protein quantification.

## 3. Results

### 3.1. Cytotoxicity of Quercetin in C2C12 Cells

We investigated the toxicity of quercetin in mouse myoblast C2C12 cells via MTT assay. Quercetin (0, 5, 50, 100, 200, 300, 400, 500 μM) was added to C2C12 cells and left to incubate for 72 h. Quercetin in amounts under 100 μM was not found to be toxic to C2C12 cells (Figure 1). Therefore, quercetin treatments below 100 μM were used to perform our experiments exploring the effects of quercetin on various skeletal muscle regeneration variables.

### 3.2. Quercetin Significantly Promoted Myoblast Fusion and Myogenic Differentiation

C2C12 cells were treated with quercetin (0, 2.5, 12.5, 25, 50 μM) in differentiation medium for 7 days in order to study the effect of quercetin on myogenic differentiation capacity. Quercetin treatment formed more myotubes than the controls (Figure 2A). Quercetin increased C2C12 cell differentiation in a concentration-dependent manner: the higher the concentration, the larger and longer the myotubes produced and the greater the multinucleation (Figure 2B). As can be seen in Figure 2C, which depicts fusion index results, quercetin (12.5 μM) increased myoblast fusion to 75% (*** *p* < 0.001) compared to controls (Figure 2C). The muscle-specific protein related to muscle cell differentiation were detected to confirm the effect of quercetin on myogenic differentiation. Quercetin (12.5 μM) had the strongest effect, increasing protein expression of MHC by 2.3 folds (** *p* < 0.01) compared to controls (Figure 2D,E). Therefore, it was decided that quercetin (12.5 μM) would be used in the remaining experiments.

### 3.3. Quercetin Did Not Affect the Exit of Myoblasts’ Cell Cycle at the Early Stage of Differentiation

Myogenic differentiation can be divided into early and late stages. We used flow cytometry to determine whether quercetin would induce C2C12 cells to exit the cell cycle at the G0/G1 phase and initiate early state differentiation. As can be seen in Figure 3, in the group treated with quercetin, there was no increase in exiting during the G0/G1 phase at 6, 12, or 24 h, compared to controls, though there was a gradual increase in the proportion of cells entering the G0/G1 phases in both groups over time, from 6 to 12 and to 24 h. Based on these findings, quercetin did not influence myogenic differentiation at the early stage by inducing myoblasts to exit the cell cycle.

### 3.4. Quercetin Significantly Enhanced Myoblast Migration

Myoblast migration is a critical factor in the promotion of myoblast alignment and fusion into multiple-nucleic myotubes. We performed a wound-healing assay and transwell migration assay to study the effect that quercetin might have on C2C12 cell migration. We observed wound closure in quercetin-treated C2C12 cells every three hours and found that quercetin increased the wound-healing ratio and enhanced C2C12 cell migration at 15 h (*** *p* < 0.001) (Figure 4A). Quercetin also increased the transwell migration ratio through the transwell membrane (Figure 4B). These findings suggest that quercetin contributes to myoblast fusion by significantly enhancing myoblast migration.

### 3.5. Quercetin Induced the Phosphorylation of AMPK

During myogenic differentiation, a certain amount of energy is required. AMPK phosphorylation (T172) has been found to regulate muscle metabolism and increase myogenic differentiation in in vitro and in vivo studies. Therefore, we wanted to investigate what effect quercetin might have on AMPK phosphorylation. As can be seen in the Western blots in Figure 5, quercetin increased the protein expression of phosphorylated AMPK (T172). This finding suggests that quercetin may influence myogenic differentiation by activating the phosphorylation of AMPK.

### 3.6. Quercetin Inhibited the Phosphorylation of PDGFR-β

We also investigated the effect of quercetin on the protein expression of PDGFR-β in C2C12 myotubes. As is shown in Figure 6, Western blot revealed that quercetin increased the protein expression of total PDGFR-β and decreased the protein expression of phosphorylated PDGFR-β (p-PDGFR-β). These results suggest that quercetin may also induce myogenic differentiation by inhibiting phosphorylation of PDGFR-β.

### 3.7. Quercetin Induced Early Upregulation of IGF-1R, Promoted Myogenic Differentiation, and Protein Synthesis via Downstream of the IGF-1R Signaling Pathway

The IGF-1R signaling pathway plays a vital role in skeletal muscle growth because it induces myogenic differentiation and protein synthesis. During the 7 days that myotubes were forming in the culture, quercetin was found to have increased protein expression of p-IGF-1R on days 3 and 4 and to have decreased the protein expression of p-IGF-1R on day 7 (Figure 7A,B). These results suggest quercetin could be used to upregulate p-IGF-1R during early myotube formation. Akt phosphorylation in T308 and S473 activated different downstream functions. As can be seen in our Western blot results, quercetin increased the level protein of phosphorylation of Akt (T308) and activated its downstream mTOR and 4EBP1 phosphorylation (Figure 7E,F). However, quercetin did not change to overexpression of phosphorylated p70S6K or downstream mTOR (Figure 7E,F). Although quercetin did not activate Akt phosphorylation (S473) (Figure 7G,H), affecting downstream GSK-3β/β-catenin, it did inactivate GSK-3β by increasing phosphorylation of GSK-3β (Ser9) and also increased the level of expression protein of β-catenin (Figure 7G,H). In addition, IGF-1R also regulated STAT3 phosphorylation. Western blot showed that quercetin treatment increased p-STAT3 (Figure 7C,D). These results indicate quercetin promotes myogenic differentiation through Akt/mTOR/4EBP1, GSK-3β/β-catenin, and phosphorylated STAT3. Moreover, quercetin contributed to protein synthesis via the mTOR/4EBP1 signaling pathway.

### 3.8. AKT Activation Is Required for Induction of Myogenic Differentiation by Quercetin

Because we found that quercetin increased the expression of phosphorylated AKT (T308) and activated the phosphorylation of mTOR and 4EBP1 to promote myogenic differentiation, we wanted to determine whether AKT was necessary for quercetin to exert its effect on myogenic differentiation. To achieve this, we incubated C2C12 cells with a specific AKT inhibitor (LY294002) for 7 days. As can be seen in Figure 8, blocking AKT lead to a decrease in expression of MHC and an inhibition of myogenic differentiation regardless of whether quercetin was present or not. Meanwhile, treatment with quercetin increased the protein expression of p-AKT (T308) and p-4EBP1(Thr37/46) in the cells pre-treated with the blocker LY294002 (Figure 8). Phosphorylation of the AKT (T308) consensus target was the first proposed phosphorylation of mTOR at S2448 [23,24]. However, pretreatment with LY294002 did not inhibit the phosphorylation of mTOR(S2448) (Figure 8), leaving us to infer that quercetin induced myogenic differentiation through the AKT/4EBP1 signaling pathway.

### 3.9. Quercetin Regulated Myoblast Migration through the ITGB1 Signaling Pathway

We wanted to investigate whether quercetin influences myoblast migration by regulating ITGB1. Western blot analysis revealed that quercetin upregulated the protein expression of ITGB1 and increased the phosphorylation of both FAK and paxillin (Figure 9A,B), indicating that quercetin induced myoblast migration via the ITGB1 signaling pathway.

## 4. Discussion

The United Nations reports that by 2050, about 22% of the global population will be over 60 years, with 5% being over 80 years. It has been estimated that approximately 42% of adults over 60 years old have difficulties performing daily life activities [25]. One of the most significant factors contributing to diminishing quality of life in the elderly is age-related reduction in skeletal muscle mass and strength [26]. Enhancing myogenic differentiation may be one way to reduce muscle loss, stimulate muscle growth, and promote muscle regeneration after injury in older adults [27,28]. This study found lower concentrations of quercetin (2.5, 12.5, 25, 50 μM) to induce myogenic differentiation (Figure 2B), particularly 12.5 μM, which was found to be the most beneficial dosage for muscle differentiation. The effect of quercetin on muscle differentiation can contribute to subsequent overexpression of MHC protein, which we found to improve myoblast fusion (Figure 2C,D). Our previous results showed that MHC expression initiated on day 3; however, quercetin did not change MHC expression on day 3 and day 4 and increased on day 7 (Appendix A). These results indicate that quercetin had the highest effect on inducing MHC expression on day 7.

Myogenic differentiation can be divided into two stages, early and late. Although various studies have investigated quercetin-induced cell cycle arrest, none have reported evidence of quercetin regulation of myoblast cell cycle arrest [29,30]. The results of our flow cytometry studies showed that quercetin did not increase myoblast cell cycle exit at 6, 12, and 24 h (Figure 3). To the best of our knowledge, this is the first study investigating the effect of quercetin on myoblast cell cycle exit. One previous study reported that theaflavin and vitamin D3 significantly increased C2C12 cell differentiation by modulating the cell cycle exit at the G0/G1 phase at the early stage of myogenic differentiation [25,31]. Therefore, it is possible that quercetin induces myogenic differentiation by increasing myoblast fusion and migration without influencing cell cycle exit.

We found that quercetin increased myoblast migration at 15 h, as evidenced by increases found in our wound-healing and transwell migration assays (Figure 4), both suggesting that it played a role in increasing myoblast fusion and differentiation. We did not know what mechanisms contributed to its effect on myoblast migration, though it has been found that in extracellular matrix environments, adhesion molecules and immune factor stimulation play roles in myoblast migration, differentiation, and repair after injury [32]. Additionally, we did not know what molecular mechanism underlay quercetin’s stimulation of myoblast migration. Therefore, we performed further studies finding that quercetin upregulated expression of ITGB1 via mediated activation of FAK and paxillin phosphorylation (Figure 9). Similarly, overexpression of platelet and endothelial aggregation receptor-1 (PEAR-1) has been observed to influence the expression of p-FAK and p-paxillin in skeletal muscle satellite cell migration via upregulation of ITGB1 and interaction with it [33]. One recent study also showed that SPARCL1, an ECM2, induced bovine skeletal muscle satellite cell migration and early differentiation through binding to ITBG1 and upregulation of FAK and paxillin phosphorylation [9]. During early myogenic differentiation, migration plays a key role in regulating myoblast fusion and promoting differentiation [9]. We found that quercetin induced myoblast migration through the ITGB1 signaling pathway, suggesting that quercetin exerts its effect on myoblast migration during early differentiation.

In the process of differentiation into myotubes, myoblasts increase the ATP activity level and energy requirements [17]. We found that quercetin increased the activation of p-AMPK (T172) (Figure 5). AMPK activation on phosphorylation of threonine 172 plays a role in controlling skeletal muscle metabolism and gene expression during exercise [34]. One recent study has found that quercetin regulates changes in skeletal muscle fiber from type II to type I by increasing both phosphorylated AMPK (T172) and downstream PGC-1α [35]. In addition, metformin, a drug for anti-diabetes, has the potential to protect muscle mass and strength via the activation of AMPK and regulator of the PGC-1α pathway [36]. Therefore, quercetin may contribute to myogenic differentiation and protect against loss of muscle through its upregulation of p-AMPK (T172). We found that quercetin increased ATP consumption by activating the phosphorylation of AMPK (T172) (Figure 5). Future studies may want to explore the effect of quercetin on increasing ATP synthesis and metabolism in skeletal muscle cells during myogenic differentiation.

Although we found that quercetin played a role in inducing myogenic differentiation, we were not sure how this could be explained molecularly. We decided to investigate its effect on the IGF-1R and PDGFR signaling pathways. We found that while quercetin increased the overexpression of total protein of PDGFR-β, it inhibited its phosphorylation (Figure 6). One previous investigation using a mouse C2C12 cell model to study PDGFR-β signaling during myogenic differentiation found that tyrosine phosphorylation of PDGFR- β (Tyr 751) decreased when myoblasts were differentiating into myotubes. [37]. Our findings and theirs suggest that quercetin may induce myogenic differentiation by inhibiting the phosphorylation of PDGFR-β. The mechanism through which PDGFR-β suppresses muscle cell differentiation is poorly understood [10]. Our experiments found that myoblasts had completely differentiated into myotubes after 7 days and that there was a higher expression of PDGFR-β in myotubes than in myoblasts, suggesting that quercetin increased the expression of total protein PDGFR-β [38].

IGF-1 plays a primary role in controlling skeletal muscle development and differentiation; hence, we wanted to find out if quercetin induced myogenic differentiation through IGF-1. IGF-1R has been reported to directly control the responses between the intracellular and extracellular muscle cells [39]. We found that although quercetin increased early overexpression of p-IGF-1R on day 3 and day 4, expression of p-IGF-1R had declined by day 7 (Figure 7A,B). Likewise, one recent study exploring the effectiveness of quercetin supplementation on IGF-1 levels after eccentric-induced muscle damage in humans found the effect of quercetin on IGF-1 level had decreased by day 7, as compared with placebo-treated controls [40]. Interestingly, MHC begins to express on day 5 following muscle damage [41], suggesting that quercetin could promote myogenic differentiation at the early stage by activating the phosphorylation of IGF-1R. Based on these findings, quercetin should have its greatest impact during a short time period during the early stage of myogenic differentiation following injury (Figure 7A,B).

Phosphorylated PI3K is activated when AKT (T308) is phosphorylated, and this is followed by the phosphorylation of the mTOR (Ser 2448) pathway responsible for protein synthesis and the regulation of muscle differentiation [42]. Activated phosphorylation of 70-kDa ribosomal S6 kinase (p70S6K) and eukaryotic initiation factor 4E binding protein 1 (4EBP-1) are the best characterization downstream of mTOR signaling and the regulation of both myogenic differentiation and protein synthesis [43]. This study found that quercetin increased protein expression of phosphorylated AKT (T308) (Figure 7E). We also found that quercetin increased the expression of phosphorylated mTOR (S2448) and 4EBP1 (Thr37/46). However, quercetin did not change the phosphorylation of P70S6K (Thr389) downstream from mTOR (Figure 7E). In addition, phosphorylation of P70S6K activated regulation via the phosphorylation of various kinases at the T421 and S424 [44]. Therefore, we hypothesized that quercetin could affect the phosphorylation of P70S6k at T421 or S424. In conclusion, quercetin activated the phosphorylation of mTORC1 (S2448) through its effect on phosphorylated AKT (T308), which then affects its downstream phosphorylation of 4EBP1, leading to protein synthesis and myogenic differentiation.

Phosphorylated AKT (S473) is directly responsible for inactivation of GSK-3β by the phosphorylation of GSK-3β Ser9, which occurs via the IGF-1-dependent signaling pathway to induce myogenic differentiation [45]. The IGF-1 and Wnt/β-catenin signaling pathways block GSK-3β activity during the induction of quiescent reverse cell differentiation. The Wnt pathway has been observed in vitro to inhibit GSK-3β and cause nuclear β-catenin accumulation in C2C12 myoblasts, which may interact with MyoD, enhancing its capacity to bind and activate target genes and increase myoblast differentiation [46]. These results showed that quercetin did not change expression of p-AKT(S473), while it increased p-GSK-3β (Ser 9) and β-catenin (Figure 7G).

In a previous study, AKT1 was found to be essential for muscle differentiation by prior investigations utilizing cells with targeted knockdowns or genetic deletions, and AKT1 activation has been defined as phosphorylation at T308 [47,48]. We found that quercetin increased the protein expression of p-AKT (T308) (Figure 7E); hence, it is worth investigating whether AKT activation was needed for quercetin to promote myogenic differentiation. LY294002 (AKT inhibitor) suppressed the protein expression of MHC and inhibited myogenic differentiation in both the presence and absence of quercetin, yet quercetin increased the protein expression of p-AKT (T308) and p-4EBP1(Thr37/46) in the cells pre-treated with LY294002 (Figure 8). It is hypothesized that phosphorylation of the AKT *(*T308) consensus target was the first proposed phosphorylation of mTOR at S2448 [23,24]. However, we did not find LY294002 treatment to inhibit the protein expression of phosphorylated mTOR (Figure 8). It has recently been demonstrated that AKT may indirectly control mTOR activity through IKK [24]. This study found AKT to directly phosphorylate 4EBP1. Therefore, quercetin may induce myogenic differentiation via an AKT/4EBP1 signaling pathway.

We hypothesized that quercetin could promote myogenic differentiation via the IGF-1R signaling pathway. However, quercetin decreased protein expression of IGF-1R phosphorylation on day 7, while increasing downstream signaling involving AKT/4EBP1, GSK-3β/β-catenin, and STAT3 phosphorylation (Figure 10). It was previously found that 67LR (67kD Laminin) was an EGCG cell surface receptor, a regulator of multiple pathways controlling cell proliferation or apoptosis in cancer cells [49]. The 67kD Laminin receptor has been shown to be the mechanism through which EGCG protects against H_2_O_2_-induced apoptosis in mouse vascular smooth muscle cells [50]. Therefore, we hypothesized that quercetin could bind to its receptors, affect these downstream intracellular cells, and increase the expression of MHC to promote myogenic differentiation. Quercetin cell surface receptors have not been identified, so future studies may want to search for them.

## 5. Conclusions

Aging can lead to declines in muscle function and proficiency in skeletal muscle regeneration after injury. Migration and differentiation play critical roles in skeletal muscle regeneration. We found that quercetin, a natural flavonoid, significantly stimulated myoblast migration and myogenic differentiation. Therefore, quercetin supplementation may hold some potential for the future treatment of sarcopenia, promotion of skeletal muscle regeneration, and the prevention of muscle loss among older people.

## Figures and Tables

**Figure 1 nutrients-14-04106-f001:**
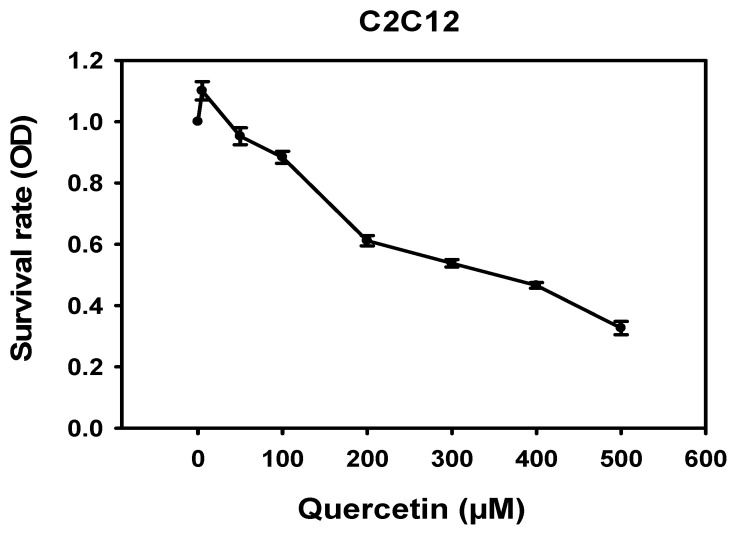
Cytotoxicity of quercetin in C2C12 cells. C2C12 cells were treated with quercetin (0, 5, 50, 100, 200, 300, 400, 500 μM) for 72 h. MTT assay was used to determine the toxicity. Survival rate diminished sharply starting at 100 μM. The line graph was created using Sigma plot software.

**Figure 2 nutrients-14-04106-f002:**
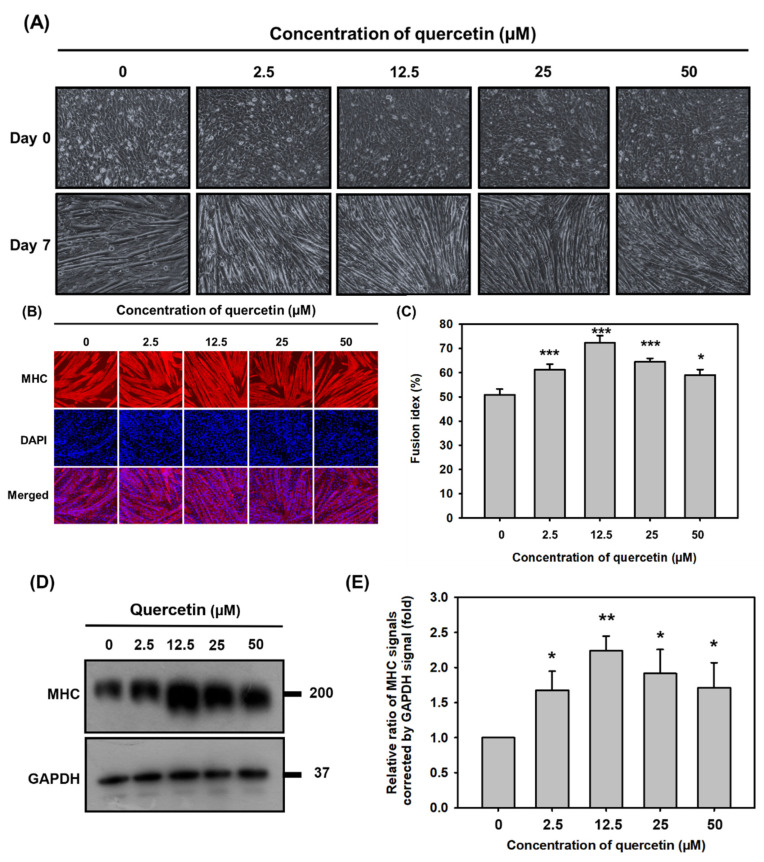
Quercetin significantly promoted myoblast fusion and myogenic differentiation. C2C12 cells were grown to 80% confluency density in a 24-well plate, in wells treated with varying amounts of quercetin (0, 2.5, 12.5, 25, 50 µM), and cultured with DMEM medium containing 2% horse serum to promote cell differentiation. (**A**) C2C12 myoblasts changed to myotubes after 7 days of quercetin treatment. (**B**) Following 7 days of differentiation, C2C12 cells were stained for MHC and DAPI to determine the fusion index (percent nuclei in MHC-positive cells). (**C**) Image J was used to calculate the fusion index, the bar chart showing the fusion index following treatment for 7 days with or without quercetin. Western blot was used to analyze the expression of MHC protein, with GAPDH used as internal control. (**D**) Western blot images and (**E**) bar graph show protein expression of MHC relative to GAPDH. Data are expressed as mean ± SD based on three independent biological repeats (*n* = 3). The differences were significant, expressed as * *p* < 0.05, ** *p* < 0.01, *** *p* < 0.001.

**Figure 3 nutrients-14-04106-f003:**
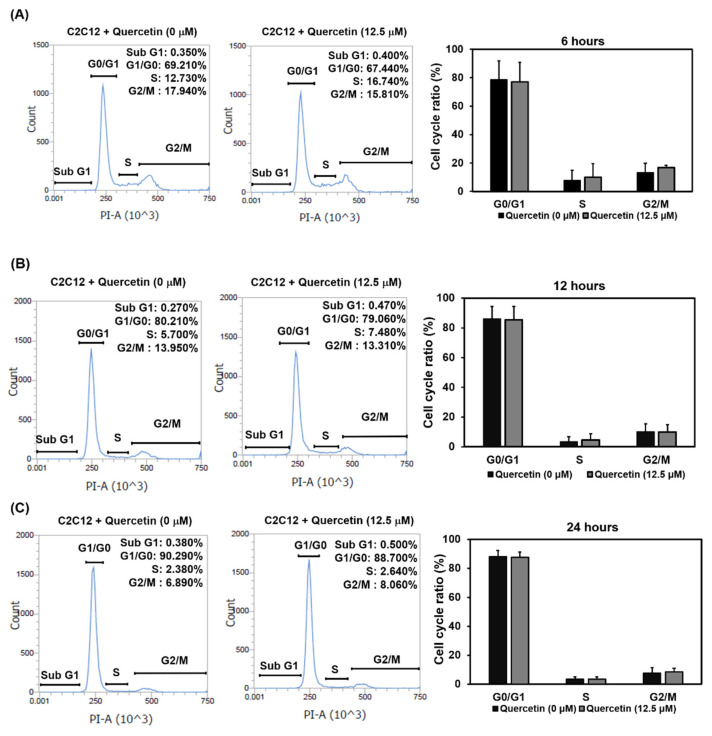
Quercetin did not affect the exit of myoblasts’ cell cycle at the early stage of differentiation. C2C12 cells were grown in a small plate to 80% confluency density, at which time the medium was replaced with a low-serum medium containing 2% FBS and 1% p/s to synchronize the cell cycle. After 24 h, C2C12 cells were treated with quercetin (0, 12.5 µM) in differentiation medium (DM) for 6, 12, and 24 h. We used Attune NxT cytometer software to study C2C12 cells’ DNA stained with PI to analyze the cell cycle. The schematic diagrams depicting our flow cytometry results show changes in the cell cycle at 6 h (**A**), 12 h (**B**), and 24 h (**C**). Data are expressed as mean ± SD based on three independent biological repeats (*n* = 3) for the 24 h treatment group and two independent biological repeats (*n* = 2) for the 6 and 12 h treatment groups.

**Figure 4 nutrients-14-04106-f004:**
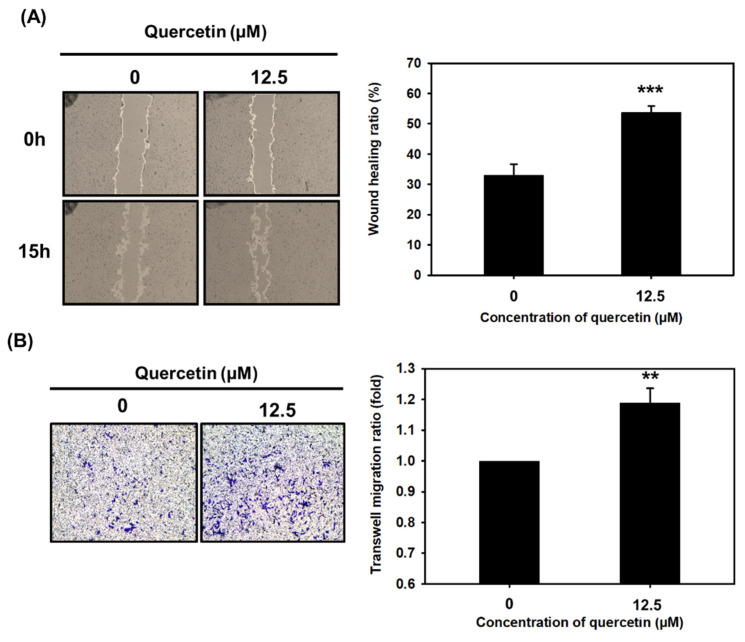
Quercetin significantly increased myoblast migration compared to controls. Wound-healing and transwell migration assays were used to study C2C12 cell migration. C2C12 cells were treated with quercetin (0, 12.5 µM) and cultured in DMEM medium containing 2% horse serum for 15 h. (**A**) Image J was used to measure the wound-healing ratio, and the bar graph shows the percentage of closure compared to controls after 15 h. (**B**) Transwell migration assay was conducted in a transwell chamber, and optical density was observed and quantified using crystal violet staining at 595 nm. Bar graph shows percentage staining. For both assays, data are expressed as mean ± SD based on three independent biological repeats (*n* = 3). The differences were significant, *** *p* < 0.001 and ** *p* < 0.01, respectively.

**Figure 5 nutrients-14-04106-f005:**
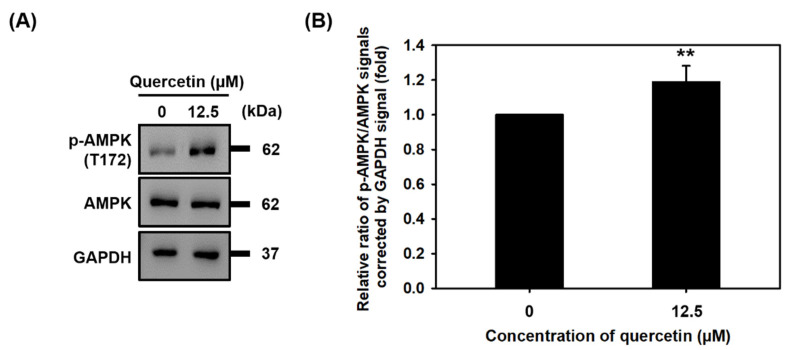
Quercetin induced the phosphorylation of AMPK. C2C12 cells were grown to 80% confluency density in a 24-well plate, each well treated with or without quercetin (0, 12.5 µM) and cultured in DMEM medium containing 2% horse serum to promote cell differentiation for 7 days. (**A**) Western blot analysis was used to study relative protein expression of AMPK and p-AMPK, with GAPDH used as internal control. (**B**) Bar graph quantitated relative protein of expression of p-AMPK and AMPK. Data are expressed as the mean ± SD based on three independent biological repeats (*n* = 3). We found a significant difference between treatment group and controls ** *p* < 0.01.

**Figure 6 nutrients-14-04106-f006:**
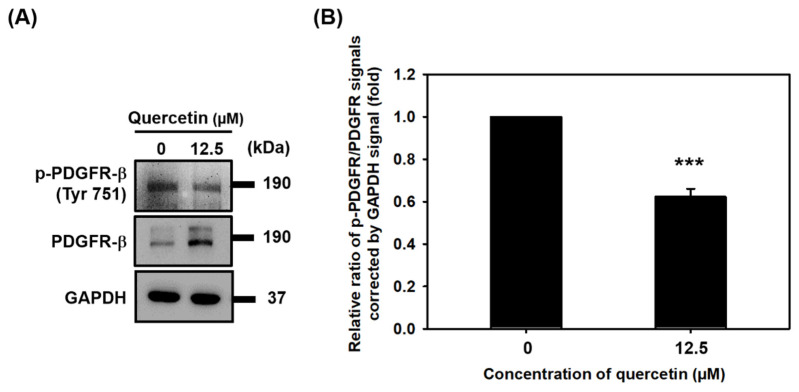
Quercetin inhibited phosphorylation of PDGFR-β. C2C12 cells were grown to 80% confluency density in a 24-well plate. The cells were untreated or treated with quercetin (0, 12.5 µM) and cultured in DMEM medium containing 2% horse serum to promote cell differentiation for 7 days. (**A**) Western blot was used to study the effect of quercetin on protein expression of p-PDGFR and PDGFR, with GAPDH used as internal control. (**B**) Bar graph shows the quantified relative protein expression of p-PDGFR and PDGFR. Data are expressed as mean ± SD based on three independent biological repeats (*n* = 3). A significant difference was found between the quercetin-treated group and controls, *** *p* < 0.001.

**Figure 7 nutrients-14-04106-f007:**
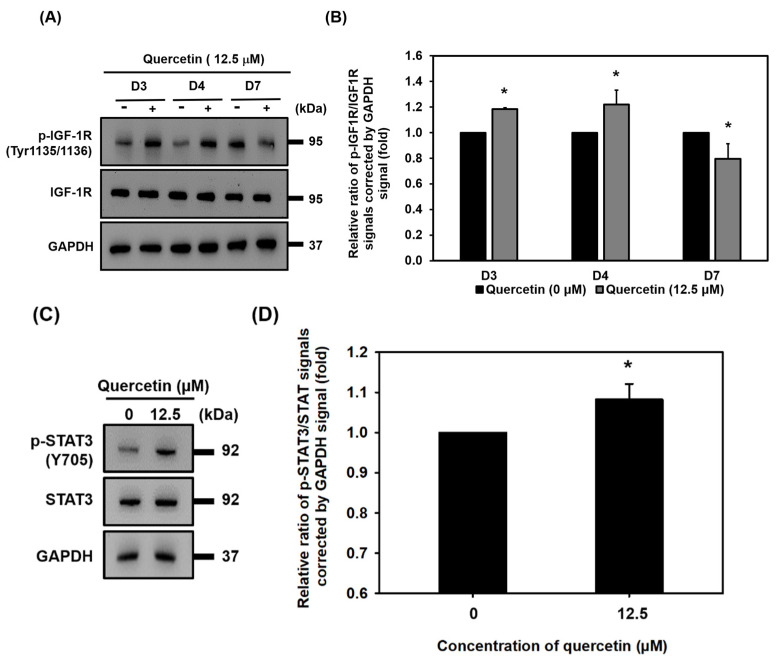
Quercetin induced early upregulation of IGF-1R and promoted myogenic differentiation and protein synthesis downstream via the IGF-1R signaling pathway. C2C12 cells were grown to 80% confluency density in a 24-well plate. The cells were untreated or treated with quercetin (0, 12.5 µM) and cultured in DMEM medium containing 2% horse serum to promote cell differentiation for 7 days. Western blot analysis was used to study the expression of proteins related to IGF-1R and AKT signaling, with GAPDH used as internal control. (**A**) Western blot images of phosphorylated IGF-1R and IGF-1R. (**B**) Bar graphs show quantified relative protein expressions. (**C**) Western blot shows analysis of protein expression of p-STAT3 and STAT3. (**D**) Bar graphs show quantified relative expressions. (**E**,**F**) Western blot images and quantification level protein related to the AKT/mTOR signaling pathway. (**G**,**H**) Western blot images and quantification level protein related to AKT/GSK-3β/β-catenin. Data are expressed as mean ± SD based on three independent biological repeats (*n* = 3), * *p* < 0.05, *** *p* < 0.001 indicate a significant difference between the control group and quercetin-treated group. (ns: not significant).

**Figure 8 nutrients-14-04106-f008:**
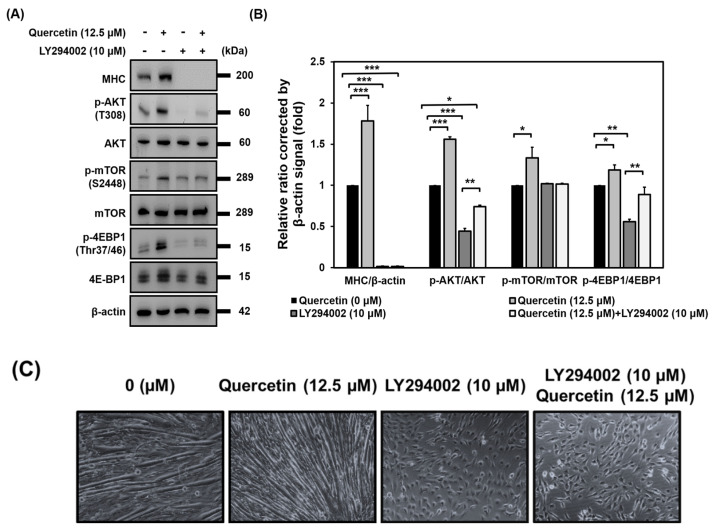
AKT activation was required for quercetin to induce myogenic differentiation. C2C12 cells were grown to 80% confluency density in a 24−well plate. The cells were pre-treated with or without LY294002 (10 µM) for 30 min. They were then untreated or treated with quercetin 0, 12.5 µM and cultured in DMEM medium containing 2% horse serum to promote cell differentiation for 7 days. (**A**) Western blot analysis was used to study the effect of quercetin on the expression of proteins related to AKT signaling, with β-actin used as internal control. (**B**) Bar graph shows the relative expression of the unphosphorylated and phosphorylated proteins. (**C**) Microscopic images of C2C12 cells represent myotube formation after 7 days. Data are expressed as mean ± SD based on three independent biological repeats (*n* = 3). There were significant differences between the controls and the quercetin-treated, LY294002-treated, and LY294002- and quercetin-treated groups, * *p* < 0.05, ** *p* < 0.01, and *** *p* < 0.001, respectively.

**Figure 9 nutrients-14-04106-f009:**
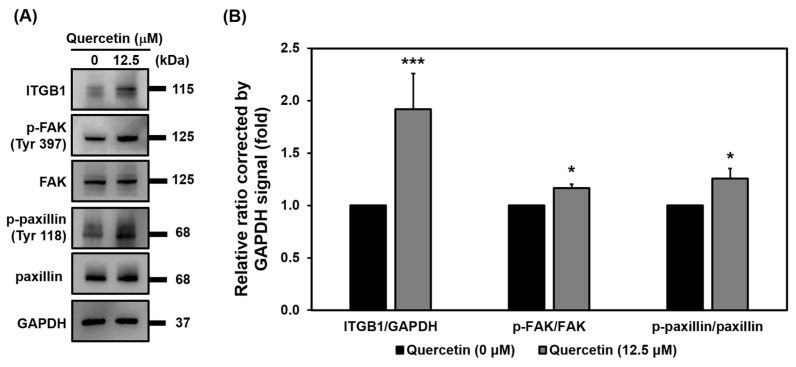
Quercetin regulated myoblast migration through the ITGB1 signaling pathway. C2C12 cells were grown to 80% confluency density in a 24-well plate. The cells were treated without or with quercetin (0, 12.5 µM) and cultured in DMEM medium containing 2% horse serum for 15 h to promote cell migration. Western blot analysis was used to investigate relative involvement of the related proteins in cell migration with GAPDH used as internal control. (**A**) Western blot images of ITGB1, p-FAK, FAK, p-paxillin, and paxillin involvement. (**B**) Bar graph shows relative protein expression of related proteins. Data are expressed as the mean ± SD of three independent biological repeats (*n* = 3). The differences between the quercetin-treated group and controls were significant, * *p* < 0.05, *** *p* < 0.001.

**Figure 10 nutrients-14-04106-f010:**
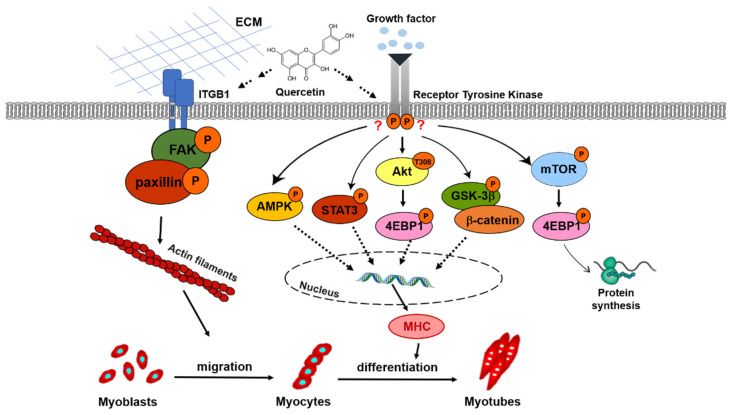
Schematic diagram of quercetin promotion of myoblast migration and myogenic differentiation. In this schema, the quercetin-activated ITGB1 receptor recruits focal adhesion proteins via a complex formed by phosphorylated FAK and paxillin, which in turn affects cell migration through interactions between the extracellular matrix and actin filaments. Quercetin induces AMPK activity, regulating skeletal muscle metabolism and increasing consumption of ATP, leading to the regulation of MHC expression to myogenic differentiation. Quercetin may indirectly affect downstream signaling involving AKT/4EBP1, GSK-3β/β-catenin, and STAT3 phosphorylation, increasing the expression of MHC and myogenic differentiation. Furthermore, quercetin also increases protein synthesis through mTOR/4EBP1 phosphorylation.

## Data Availability

The data that support the findings of this study are available from the corresponding author upon reasonable request.

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
