# Peer review of "The Promotion of Migration and Myogenic Differentiation in Skeletal Muscle Cells by Quercetin and Underlying Mechanisms"

_nutrients, 2022, doi:10.3390/nu14194106_

Round 1
Reviewer 1 Report
The paper is mainly well written and fit to the international standards. However some missing data should be given for the acceptance. It would be nice to complement the experimental set to show how MHC expression is dependent on days in ctr and quercetin treated (12.5 uM) samples. A more important missing experiment is to show how LY affect the differentiation process (by showing morphology) and whether quercetin can overcome LY effect.
Cytotoxicity of quercetin, Fig 1
In the text 8 measured point is given, but on the graph only 7 is presented, probably data for 5 uM point is missing. PLease correct this.
To 3.2, Fig 2
It seems based on 2B that MHC expression is quite high even at day 0. It would be more informative to present western or FACS analysis about MHC expression in dependence of days at least for the ctr, not only at the endpoint (as in D). Scale bar is missing on microscopy pictures, please supply it.
To 3.5, Fig 5
ALthough it can not be assumed based on a pdf presentation, the western showed in Fig 5/A shows a marked difference in pAMPK signals but in quantification based on 3 blots the difference is only 20%. My personal experience is that a 20% difference in western signals can hardly be seen by eyes - but of course 5/A is only an example, picture may be missleading. But anyway if conclusion is drawn from Fig5B it means that the difference of ctr and quercetin treated sample is very faint. I think, in general, at least a 1.5 times difference means a real difference in western quantification.
To Fig 6.
PDGFR-b antibody labelled 2 bands/lane - you should specify which was measured and it would be good to know what is the second lane - a non-specific band or a variant form of the protein.
To Fig 7.
In the text it is said, that "quercetin was found to have increased protein expression of 358 p-IGF-1R on days 3 and 4 and to have decreased the protein expression of p-IGF-1R on 359 day 7" (line 358-359). Its a common fault to write protein expression instead of activation (by phosphorylation). As I can see from pictures expression level of IGFR has not been changed only its active phosphorylated form. PLease correct these terms everywhere.
Here also the picture A seems to show higher difference than quantification (7B)
To Fig 8.
It is mentioned that LY inhibited MHC expression and myogenic differentiation independently of the presence of quercetin, but you should present data about how LY294002 modified the differentiation process (migration or fusion index) and whether quercetin have any effect on morphology in this case. Conclusion that quercetin induced myogenic differentiation goes through AKT/4EBP1 signalling would be proven only if there would be a significant differentiation due to quercetin treatment even in the presence of LY.
In all western pictures molecular weight markers are missing, please give these.
In all figures "relative to protein expression of X/Y (fold) is not a clear definition, rather (Band intensity) Relative Ratio of X/Y signals corrected by GAPDH signal would be perfect.
Regarding methods:
It is not mentioned what was the final DMSO concentrations in samples, and whether control samples contained this amount also.
To conclusion:
It is correct, also it should be mentioned that quercetin changed only a little the phosphorylation levels of various signalling molecules in the applied circumstances. ANd it seems that several signalling pathways can be involved in its differentiation effect. Unfortunately your study does not apply positive controls for stimulating the given signalling routes like IGF1 for IGFR related ways.
Reviewer 2 Report
The study assessed the influence of quercetin on migration and myogenic differentiation in a muscle cell culture.
Abstract, lines 32-33: “The molecular mechanisms of quercetin promote myogenic differentiation investigated via activated transcription factors STAT3 and AKT signaling pathway.” This sentence requires some re-wording. Perhaps try “The molecular mechanisms of quercetin include the promotion of myogenic differentiation via activated transcription factors STAT3 and the AKT signaling pathway”
In the abstract, please add that the cell lines were from mice.
Lines 107 and 518: Change “The previous study…” to “A previous study…”
Line 113: Here you have listed tea as a vegetable or fruit, which is not accurate. I suggest re-arranging the wording in this sentence.
The introduction section is very long. I suggest you restrict this to a discussion of the pathways and mechanisms which were evaluated in your study.
Statistics: Simple t-tests are used to assess the data; however, there are multiple factors with each analysis (or multiple levels) – for example, different concentrations of quercetin or multiple time points of incubation. The data would be better analyzed using an ANOVA with factors of condition (quercetin and control) and time.
Quercetin was assessed in the cell cultures at a concentration of 12.5 µM. Is it feasible to use a dosage of quercetin high enough to achieve these concentrations in living systems?
Figure 10: This is a nice figure, but can you increase the resolution in the figure?
Round 2
Reviewer 1 Report
I accept the answers. Only a minor point - You write in the answer that C2C12 cells require 7 days to differentiate, I think this is an important and non trivial information that was not mentioned in the paper. It would be informative to write it in the paper as well. Scale bars still missing from microscopy pictures (although it has no great importance in this case).